# Genetic Diversity, Analysis of Some Agro-Morphological and Quality Traits and Utilization of Plant Resources of Alfalfa

**DOI:** 10.3390/genes13091521

**Published:** 2022-08-24

**Authors:** Mervat R. I. Sayed, Khalid S. Alshallash, Fatmah Ahmed Safhi, Aishah Alatawi, Salha Mesfer ALshamrani, Eldessoky S. Dessoky, Ashwaq T. Althobaiti, Mohammed M. Althaqafi, Hany S. Gharib, Wafaa W. M. Shafie, Mamdouh M. A. Awad-Allah, Fadia M. Sultan

**Affiliations:** 1Forage Crop Research, Section, Field Crops Research Institute, Agricultural Research Center, Giza 12619, Egypt; 2College of Science and Humanities-Huraymila, Imam Mohammed Bin Saud Islamic University (IMSIU), Riyadh 11432, Saudi Arabia; 3Department of Biology, College of Science, Princess Nourah bint Abdulrahman University, Riyadh 11671, Saudi Arabia; 4Biology Department, Faculty of Science, University of Tabuk, Tabuk 71421, Saudi Arabia; 5Department of Biology, College of Science, University of Jeddah, Jeddah 2795, Saudi Arabia; 6Department of Biology, College of Science, Taif University, Taif 21944, Saudi Arabia; 7Department of Biotechnology, Faculty of Science, Taif University, Taif 21974, Saudi Arabia; 8Department of Agronomy, Faculty of Agriculture, University of Kafrelsheikh, Kafrelsheikh 33516, Egypt; 9Central Lab. For Design and Stat. Analysis Agricultural Research Center, Giza 12619, Egypt; 10Rice Research Department, Field Crops Research Institute, Agricultural Research Center, Giza 12619, Egypt

**Keywords:** perennial forage crops, forage yield, genetic parameters, crude protein percentage and crude fiber percentage

## Abstract

Alfalfa (*Medicago sativa* L.) is one of the most important perennial forage crops to build effective diets for livestock producers. Forage crop improvement depends largely on the availability of diverse germplasms and their efficient utilization. The present investigation was conducted at Ismailia Agricultural Research Station to assess twenty-one alfalfa genotypes for yield components, forage yield and quality traits during 2019/2020 and 2020/2021. The genotypes were evaluated in field experiments with three replicates and a randomized complete block design, using analysis of variance, estimate of genetic variability, estimate of broad sense heritability (h_b_^2^) and cluster analysis to identify the inter relationships among the studied genotypes as well as principal component analysis (PCA) to explain the majority of the total variation. Significant differences were found among genotypes for all studied traits. The general mean of the studied traits was higher in the second year than the first year. Moreover, the combined analysis showed highly significant differences between the two years, genotypes and the year × gen. interaction for the traits studied. The genotype F18 recorded the highest values for plant height, number of tiller/m^2^, total fresh yield and total dry yield, while, the genotype F49 ranked first for leaf/stem ratio. The results showed highly significant variation among the studied genotypes for crude protein %, crude fiber % and ash %. Data revealed that the genotypes P13 and P5 showed the highest values for crude protein %, whereas, the genotype F18 recorded the highest values for crude fiber % and ash content. The results revealed high estimates of genotypic coefficient and phenotypic coefficient of variation (GCV% and PCV%) with high hb2, indicating the presence of genetic variability and effective potential selection for these traits. The cluster analysis exhibited considerable genetic diversity among the genotypes, which classified the twenty one genotypes of alfalfa into five sub-clusters. The genotypes F18, F49, K75, S35, P20, P5 and P13 recorded the highest values for all studied traits compared with other clusters. Furthermore, the PC analysis grouped the studied genotypes into groups and remained scattered in all four quadrants based on all studied traits. Ultimately, superior genotypes were identified can be utilized for crop improvement in future breeding schemes.

## 1. Introduction

Forage legumes are important for farming systems, especially as protein sources, which are usually the most limiting nutrients in animal diets and can be grazed, harvested and fed fresh or stored as hay or silage. In general Alfalfa (*Medicago sativa* L.) is already the most cultivated species in the world for forage production. Alfalfa is a perennial forage legume known for its high forage quality and positive effects on soil fertility [1]. Its economic significance is based on its high potential for production of biomass, and it is reported to withstand long periods of water deficit by impeding its vegetative growth. In addition, it has access to water from the depths through its deeper root system [2]. Alfalfa (*Medicago sativa* L.) is a major forage crop grown in more than 80 countries on every continent of the world in an area of more than 35 million hectares. It is grown in a wide range of climatic and compositional conditions ranging from semi-arid to humid regions.

In Egypt, the regional varieties have been planted in oases for many years. Currently there are nearly 90,000 acres of alfalfa grown annually and the demand for the crop continues to rise [3]. Yield data shown an approximate 18% increase in the area allocated to alfalfa recorded from the growing season 2014/2015 to 2016/2017, however, the increase in production was only 7%, indicating the need for improving crop productivity. Because alfalfa adapts to conditions of excessive heat, drought and salinity, it is preferred over the berseem reclamation in Nubaria, Ismailia, in the 1.5 million feddan reclamation project launched by the Egyptian government in 2014. The need for new varieties with higher yields and better feed quality is essential to deal with the increased demand for the crop. Regarding to its importance as a forage crops, scientific efforts have been and are being devoted annually in breeding program regarding the improvement of both yield and quality of alfalfa. Variability for agronomic and morphological characteristics of alfalfa are frequently used in breeding programs for developing cultivars with high forage production and quality [4]. The success of a good breeding and selection program usually depends on the genetic variability present in the breeding materials and the variation in the population and helps to understand genomic composition, identify genes for vital traits and conserve and classify genetic variation in the plant germplasm.

Seiam and Mohamed (2020) [5] found highly significant differences in years and seasons for fresh yield, dry yield and protein content of twenty five genotypes of alfalfa. In addition, phenotypic and genotypic coefficients of variation and heritability are very important indicators in improving traits [6]. Hamd Alla et al. (2013) [7] showed the values of phenotypic and genotypic coefficients of variation for yield and yield components and revealed relative variation among the tested varieties which were less influenced by the environment.

One of the main goals in alfalfa breeding is the improvement of forage quality through increasing protein and fiber contents that are closely associated with the relative production of the leaf and stem components as measured by leafiness percentage, but more often as leaf/stem ratio (LSR) and the nutritive value of the stem [8]. Moreover, curd protein is considered an important indicator of forage quality [9].

Several studies have shown the importance of selection and assessment varieties for quantitative and yield ability in any breeding program, so that the varieties can be introduced to a given local environment [10]. In addition, multivariate statistical methods and numerical classifications are used, such as cluster analysis which is an effective tool aimed at determining the degree of genetic variation between tested genotypes based on their performance and their contributing characteristics [11] and Principal Component Analysis (PCA), which has been widely used in the studies of variability in germplasm collections of many species [12]. The current study was conducted to study and achieve the following objectives: (i) to evaluate twenty-one alfalfa genotypes by focusing on genetic diversity and heritability for yield components, forage yield and quality traits; (ii) to To determine genetic relationships among tested genotypes using cluster analysis and principal component analysis.

## 2. Materials and Methods

### 2.1. Plant Material

Twenty-one alfalfa genotypes of different geographical origin in Egypt were collected and evaluated in this study (Table 1). The research was conducted on sand soil, Davis ROE, Bennett HH (1927), Soil Survey Staff 1960 and Baillie, I. C. (2001) [13,14,15], Soil particle size distribution of the experimental site is shown in Table 2, at the Agricultural Research Station Farm in Ismailia Governorate, Egypt. The farm is located at 30°35′41.9″ N latitude and 32°16′45.8″ E longitude and the experiments were conducted in 2019/2020 and 2020/2021.

### 2.2. Experimental Design

On 28 October 2019 season, the genotypes were separately sown in a randomized complete block design (RCBD) with three replications. A plot with an area of 6 m^2^ for each genotype was seeded at the rate of 20 kg/fed (48 kg/ha). Seeds were inoculated with Rhizobium melolitii prior to seeding. Plots received 60 kg/fed (360 kg/ha) of P_2_O_5_, 96 kg/fed (120 kg/ha) of K_2_O before emergence and 20 kg/fed (48 kg/ha) nitrogen after emergence. The first irrigation was applied seven days after sowing. The following irrigations applied every ten days in summer and every fifteen days in winter. The first cut was taken in 11 January 2020 and then nine cuts were taken on each season (2020 and 2021).

### 2.3. Data Collection and Statistical Analysis

During the two-year investigation, five phenotypic traits were studied in this research. The following:A.Agro-morphological traits:1-Plant height (cm): Five plants were measured for each plot at each cut and averages were calculated for all the cuts.2-Number of tillers/m^2^.3-Leaf/stem ratio: Leaves of alfalfa plant samples were separated from stems then leaves and stem samples were oven dried at 70 °C for 72 h untill constant weight. Then the dry separated leaves and stems were weighted and was calculated was calculated by dividing the weight of dry leaves by the weight of dry stem.4-Total fresh yield: During each cut, fresh yield was weighted after harvesting the complete plot (kg) and convert to ton/fed (fed = 4200 m^2^). Total fresh yield was calculated by sum of cuts yield.5-Total dry yield: Total dry matter was estimated by calculating the percentage of dry matter multiplied by the fresh yield. This was conducted by cutting the green plants manually and weighing them on a digital scale, then placing them in a shad for drying and transferring them to an electric oven at a temperature of 105 °C for a period of time until a stable weight was reached.
B.Chemical Composition:


The chemical composition was determined according to the conventional method recommended by the Association of Official Agricultural Chemists (A.O.A.C) [16] on the dried samples at 70 °C for each cut of the second season only to determine:
1-Crude protein (CP): The N-contents of the sample was determined by [16] and the recorded value of nitrogen was then multiplied by 6.25 according to [17].2-Crude fiber (CF) contents were determined according to the methods described by [18].3-Ash (%) was estimated by multiplying forage dry yield/fresh forage × 100.

### 2.4. Statistical Analysis

The collected data were subjected to individual and combined analysis of variance (ANOVA) of randomized complete block design for each one of the two seasons and across them [19]. Bartlett test of variance homogeneity was carried out before the combined analysis. to detect the significant differences among genotypes means at 0.05 and 0.01 probability level was used for the least significant difference (LSD) test. The genotypic (GCV %) and phenotypic (PCV %) coefficients of variations were estimated using the pertinent mean square expectations according to the method suggested by [20,21] and broad sense heritability (hb2) was estimated as described by [22].

The GCV and PCV values were classified as follows: low = 0–10%; moderate = 10–20%; and high ≥ 20%, according to [23].

Broad-sense heritability was categorized as follows: low = 0–30%; medium = 30–60%; and high = above 60%, according to [20].

The hierarchical cluster analysis was performing on the standardized data using a measure of Euclidean distance and Ward minimum variance method as outlined by [24], while principal component analysis (PCA) was carried out as explained by [25].

## 3. Results and Discussion

A.Agronomic Traits:
1.Variability in Agronomic Traits


Data in Table 3 and Table 4 show the mean squares for plant height, number of tiller/m^2^, leaf/stem ratio, total fresh and dry yields (ton/fed) of twenty-one alfalfa genotypes in two successive seasons (Table 3) and the combined across the two years, Table 4. The results indicated significant and highly significant differences among alfalfa genotypes for all traits under this study. Moreover, from the analysis of variance for combined the results showed significant and highly significant differences between seasons and among genotypes for all studied traits, Table 4. In addition, the year × genotypes interaction was significant for all traits under this study. These results are confirmed with [26,27] who reported significant differences among genotypes (nine cultivars of alfalfa) as well as between genotypes and years for plant height, number of tillers, leaf/stem ratio, fresh and dry forage yields under the New Valley environment, Egypt. Significant differences among alfalfa cultivars and populations for plant height, leaf/stem ratio and green mass yield were detected by [5,28].

In addition, Jia et al. (2018) [29] revealed that highly significant differences among 75 genotypes of alfalfa for different phenotypic traits were observed in both years. At the same time, there were highly significant variations among the genotypes for plant height, no. of tillers, leaf to stem ratio and fresh weight in ten genotypes of alfalfa, [26]. These results reflect the different genetic backgrounds of the studied genotypes to help the breeder in selecting the germplasm collections for using in breeding programs. Therefore, it is recommended to use a broad genetic base from a variety of sources to include most of the genetic determinants of these traits [26,30].

The mean performance of forage yield and its components are tabulated in Appendix A. The data revealed a high amount of variation in studied traits for the studied genotypes, where by the second year had higher values for agronomic traits. That is consistent with the results reported by Sabanci et al. (2013) [31]. Data revealed that F18 recorded the maximum value for plant height (60.78 and 58.62 cm), number of tiller/m^2^ (335.23 and 326.34), total fresh yield (61.03 and 58.36 ton/fed) and total dry yield (15.26 and 13.58 ton/fed) in both years, respectively, while, F49 ranked first (1.859 and 1.644) followed by F18 (1.785 and 1.571) for leaf/stem ratio in both years, respectively. Otherwise, the genotype S4 had the lowest value for all studied traits except for number of tiller/m^2^, where it recorded (43.43 and 40.55 cm) for plant height, (1.016 and 1.231) for leaf/stem ratio, (36.26 and 34.45 ton/ fed) for total fresh yield and (7.91 and 6.62 ton/fed) for total dry yield in both years, respectively.

The combined data analysis over the two years revealed that F18 had the largest values for plant height (59.70 cm), number of tiller/m^2^ (330.79), total fresh yield (59.70 ton/fed) and total dry yield (14.42 ton/fed) than the other genotypes, while, F49 ranked first for leaf/stem ratio (1.752), (Appendix A) and Figure 1, Figure 2, Figure 3, Figure 4 and Figure 5. The results showed the superiority of some lines and genotypes (F18, F49, K75, P5, P13, P20, S35 and F28) in the total dry yield (ton/fed) and the total fresh yield (ton/fed), as their production of total fresh yield reached 50 tons or more per feddan under the conditions of the experiment for these lines, although these genotypes were selected from different climatical and environmental regions (southern Egypt, where the temperature is very high and the conditions of the continental or desert climate), which indicates that they excel under a wide and varied range of climatic and environmental conditions. Thus, these lines (genotypes) can be used as commercial varieties in the event of their continued superiority after further evaluation and more study, in addition to that they can be used in programs to improve and develop new lines and varieties of alfalfa that are tolerant to climate change conditions in Egypt. The results showed the presence of diversity in alfalfa at the phenotypic level using morphological and yield related attributes, which conformed that the presence of genetic diversity among alfalfa genotypes can be exploited for future crop impartment breeding programs. These results are in harmony with those reported by [32], that the investigated six alfalfa genotypes exhibited high total variability for plant height and forage yield. Some previous studies showed the broadest range of variation for some alfalfa cultivars in all morphological traits evaluated [33,34]. Generally, the alfalfa forage yield depends on plant height, and number of tillers. Therefore, these traits are the most considerably used selection standards in alfalfa breeding programs, [35]. Previous studies showed that plant height positively correlated with production in alfalfa [36]. Therefore, a standard is used when choosing the best genotypes for crossing and breeding programs as well as selected better genotypes and predicting high production during alfalfa growing and breeding [37]. The leaf/stem ratio, the fresh yield and dry yield are an important indicator for forage palatability assessment and affects the forage production. The results of this study pointed to F18 and F49 as having higher plant height, leaf/stem ratio, fresh yield and dry yield. Therefore, these genotypes are suitable to improve promising new lines for high yield.

2.Genetic parameters:

Noteworthly differences were observed for estimates of the agronomic the studied traits, Table 5. The results revealed that the estimates of the phenotypic coefficient of variation (PCV) were higher than the genotypic coefficient of variation (GCV) for all studied traits, signifying that the apparent variation is not only due to the genetic effects, but also due to environmental effects. However, the differences between PCV and GCV for most of the traits were small, indicating high probabilities of genetic progress through selection under this study, whereas, the environmental impact on any trait is indicated by magnitude of the differences between the genotypic coefficients and the phenotypic coefficients of variance. Whereas large differences reflect a large environmental influence, while small differences show a high genetic influence, and these findings conformed with to the findings of [5,7,38]. The results of the first year showed that, the highest genotypic and phenotypic coefficients of variation were recorded for leaf/stem ratio (22.26 and 23.03%), followed by total dry yield (15.48 and 16.29%), respectively. However, the lowest genotypic and phenotypic coefficients of variation were recorded for number of tiller/m^2^ (0.56 and 0.62%), respectively. The heritability (H2%) of differences among genotypes were high. The values of heritability were ranged from 87.45 for number of tiller/m^2^ to 95.22 for total fresh yield. Generally, the heritability estimates were high for all studied characters. The high heritability suggested that selection of these traits would be efficient, being less affected by environmental influences [39]. Heritability considered a measurement for phenotypic variance attributable to genetic a cause that has prophetic function in plant breeding [40].

On the other hand the results of the second year revealed that, the highest genotypic and phenotypic coefficients of variation were recorded for leaf/stem ratio (24.10 and 25.14%), followed by total dry yield (14.19 and 15.14%) and total fresh yield (10.35 and 10.63%) as moderate values. Plant height (8.06 and 8.36%) and leaf/stem ratio (0.50 and 0.54%), were recorded low values of genotypic and phenotypic coefficients of variation, while, the estimates of heritability in broad-sense were generally high for all studied characteristics and recorded values from 86.93% for number of tiller/m^2^ to 95.45 for total fresh yield.

The genetic estimates of the combined data across the two years showed large differences for all studied traits. The estimates of genotypic and phenotypic coefficients of variability were moderate for leaf /stem ratio (14.53 and 16.69%), total fresh yield (11.67 and 12.43%) and total dry yield (10.64 and 11.82%), while, the genotypic and phenotypic coefficients of variability were low for plant height (8.91 and 9.31%) and number of tiller/m^2^ (1.43 and 1.51%). Heritability estimates were high for all studied characters and with the range values from (87.06%) for leaf/stem ratio to (95.70%) for plant height. These results are in accordance with the finding of [41] who found low to moderate estimates of genotypic and phenotypic coefficients of variation (GCV and PCV) with high heritability for agro-morphological traits of alfalfa. However, the high estimates of heritability indicate the prevalence of genetic effects which are less affected by the environment. This finding is in agreement with results obtained by [42]. Thus, estimating heritability helps breeders to determine resources efficaciously to select desirable traits and to maximize genetic gains with little time and resources [43].

B.Chemical Composition:
1-Variability in Agronomic Traits


The analysis of variance indicated significant and highly significant differences among the genotypes of alfalfa for crude protein (CP %), crude fiber (CF %) and ash % in the first growing year, Table 6. These results are in line with those obtained by [5,44,45] who found highly significant differences among the alfalfa population for crude protein, crude fiber and ash %.

Highly significant differences were found among the investigated genotypes for different quality traits in the first year, Figure 6, Figure 7 and Figure 8 and Appendix A. The value of crude protein content of alfalfa genotypes ranged from (17.11–26.96%), crude fiber content ranged from (21.64–31.46%) and ash% content ranged from (6.11–13.98%) in the first year. Data revealed that P13 ranked first (26.96%) followed by P5 (24.87%) for crude protein%, while, F18 recorded the highest values (31.46% and 13.98%) for crude fiber% and ash%, respectively. On the contrary, the lowest value was recorded with S4 in all quality traits under study in the first year. These results are in agreement with those reported by [28,46,47]. The most useful forage quality traits include crude protein%, crude fiber% and ash% [48] to raise the nutritional value of alfalfa for livestock [49]. Therefore, the main goal in alfalfa breeding for the improvement of forage quality is increasing protein%, fiber% and ash content. Variations were found among eight alfalfa cultivars in CP% over two years, during a study of the nutritive value of alfalfa [47]. The ratio of leaf/stem ratio directly reflects the nutritional value of the forage, as a large quantity of leaves in the forage indicates its good quality. In the same context, several studies showed the relationship between the leaf/stem ratio and quality. The forage quality can be improved by selection for increased leaf/stem ratio in alfalfa cultivars, [7,50]. In general, forage yield and quality are complex traits whose pressure is influenced by plant genetics as well as environmental factors [51]. Therefore, it is necessary to understand the relationships between quality and agricultural traits that may help in breeding programs [52].

2-Genetic parameters:

Data in Table 7 revealed that the phenotypic and genotypic coefficients of variability were high for crude fiber% (21.64 and 23.56%) and ash% (20.04 and 21.42%), while they were moderate for crude protein% (12.52 and 14.15%), respectively. This indicates the presence of exploitable genetic variability for these traits, similar results were obtained by [53]. Therefore, the selection will be useful for the development of traits related to the degree of desired variance [54]. Heritability (H2%) estimates were generally high for all quality traits and recorded values from 88.48% for crude protein% to 93.56% for ash%, (Table 7). In general, all quality traits had high heritable variation. Hence, it can be assumed that mainly by phenotypes the genotypes of almost all traits are determined, [44]. Therefore, the selection of traits with high heritability values accelerates the improvement of traits less influenced by environmental factors [55].


**Cluster analysis:**


The neighbor-joining dendrogram is a tool for classifying objects, which has been widely used as an effective method to discover the structural associations among tested genotypes and provides a hierarchical classification of them. The genetic divergence can providing a visual idea about variability existing in the twenty–one genotypes, in addition to pledge the continued genetic improvement. In this study, based on Euclidean distance utilizing the UPGMA, the tested genotypes were estimated with agro-morphological and quality traits, and distance was achieved as revealed in dendrogram graph (Figure 9). The studied genotypes were presented in groups based on the results of the cluster analysis to infer relationships among genotypes (Table 8).

The genotypes of each cluster varied from 1 to 5. Cluster 1 contained 4 genotypes, Clusters 2 and 3 contained 5 genotypes, Cluster 4 contained one genotype only and Cluster 5 contained 6 genotypes. From the data in Table 8 and dendrogram (Figure 9), the aimed genotypes were grouped into two main clusters, that is; Cluster A and B. Nevertheless, the first main cluster was divided into three sub-clusters which could be named, 1, 2 and 3. The sub-cluster number one consisted of four genotypes (D32, D89, M3 and S12). The second sub-cluster included five genotypes (F11, K42, F26, M9 and S4). The third sub-cluster included five genotypes (D47, F28, D95, S5 and S6). The second main cluster consisted of two sub-clusters (4 and 5). The fourth sub cluster was comprised of one genotype (F18). While the fifth sub cluster included six genotypes (F49, K75, S35, P20, P5 and P13). These results are in agreement with [26,56]. The cluster analysis was used to distinguish diversity based on morphological and physiological characteristics among genotypes, [56]. The genetic diversity between the genotypes of plant height, no. of tillers, leaf to stem ratio and fresh weight in ten genotypes of alfalfa were reported by Riasat et al. (2021) [26], who, classified ten alfalfa genotypes into 4 clusters on the basis of morphology and yield by cluster analysis. Dendrograms were constructed using morpho-physiological and biochemical traits for *Medicago truncatula* [12] and with morphological traits for Pinus koraiensis [57]. Generally, the genotypes S5 with S6 and P5 with P13 were more closely correlated to each other, whereas less similarity was found between D32 and P13 genotypes.


**Principal Component Analysis:**


The principal component analysis (PCA) grouped the studied genotypes into groups and remained spread in all four quadrants based on the yield components, forage yield and quality traits. The classification of groups depends on the material involved under study [58]. The genotypes F28, F18, S4 and P13 were placed at extreme positions from the origin in the PCA biplot whereas the genotypes F11, K42 and D89 were concentrated around the origin on PC2. The results of the PCA analysis were presented in groups of genotypes to infer relationships among genotypes (Figure 10).

These results are similar to the results obtained by [59,60] who studied genetic diversity and variability of forage crops by principal component analysis using agronomic characteristics.

The principal component analysis (PCA) was useful to identify the characters which were the main source of the variability and to clarify the genetic diversity in genotypes and could calculate each principal component contribution and the accumulative effect. In addition, it has been widely used in studies of variance in germplasm collections of many species, [26,61,62]. The first five principal components gave eigenvalues greater than 1.0 and explained 88.50% of the total variability among the genotypes for all the studied traits, (Table 9). Riasat et al., (2021) [26] reported an assessment of genetic diversity for ten alfalfa genotypes at the phenotypic and genotypic level and found that the first five PCs contributed 90.19% of the entire variability.

The first PCA, which is the most important component, accounted for 5.76 of the eigen value and 53.24% of the total variability and various traits viz., plant height, number of tillers/m^2^, leaf/stem ratio, total fresh yield, total dry yield and quality traits contributed significantly towards variation. Total fresh yield had the maximum contribution followed by plant height in PC1. Similar results were obtained by [63] who studied morphological diversity of wild genetic resources of alfalfa and detected that the first PC explained 56.4% of the total variability in the measured traits and was associated with biomass production, which is approximately congruent with our results. However, Annicchiarico (2006) [64] revealed that the first PC included only 35% of the total variability which is lower than our results. Perhaps this variance was linked to a higher level of homogeneity between the alfalfa materials under study. PC2 has an eigen value of 3.11 and variance contribution rate of 15.73%, and plant height, crude fiber% and ash% contributed positively but the other traits contributed negatively to variation in PC2. These results are line with [65] revealed that the second PC included for 16.24% of the variability. PC3 contributed 13.21% in total variation. Leaf/stem ratio was a significantly positive contributor while total dry yield was a negative contributor for PC3. In the remaining PCs, number of tiller/m^2^, crude protein% and ash% were the variables responsible for variation. Similar results were obtained by [65], who reported that the PCA separated the majority of the twenty-seven cultivars and populations of alfalfa for thirteen traits into the first three principal components and the first PC ranked 58.21% of the total variability. In addition, Tlahig et al., (2017) [66] reported that the PCA separated the majority of the thirty-nine genotypes of alfalfa for six traits into the first six principal components and the first PC ranked 34.45% of the total variability. The results obtained from the cluster analysis are similar to the results of PCA. Therefore, PCA is preferred to be used in conjunction with the dendrogram to gain a better understanding of the relationships between genotypes, [67]. The crossing of parents with greater inter-cluster distances could produce desirable recombinants, while crossing parents from lower inter cluster distances seems to not produce desirable recombinants [68].

## 4. Conclusions

The results of this study revealed a wide phenotypic variability for most of the studied traits in the twenty-one genotypes of alfalfa. The genetic estimates showed highly significant differences. The cluster analysis divided into two major clusters and the PCA separated the majority of genotypes into the first five principal components. Generally, from the results of this study, it could be concluded that, selection in this genotypes is good to improve these traits, and also the genotypes F18, F49, K75, S35, P20, P5 and P13 showed the highest values in the total dry yield (ton/fed) and the total fresh yield (ton/fed). These genotypes were selected from climatically and environmentally hard regions where the temperature is very high with the conditions of the continental or desert climate, this indicated that these excel under a wide range of climatic conditions. Thus, these lines can be used in a commercial variety in the event of its continued superiority, in addition to that it can be used in programs to improve and develop new lines and varieties of alfalfa that are tolerant to climate change conditions in Egypt.

## Figures and Tables

**Figure 1 genes-13-01521-f001:**
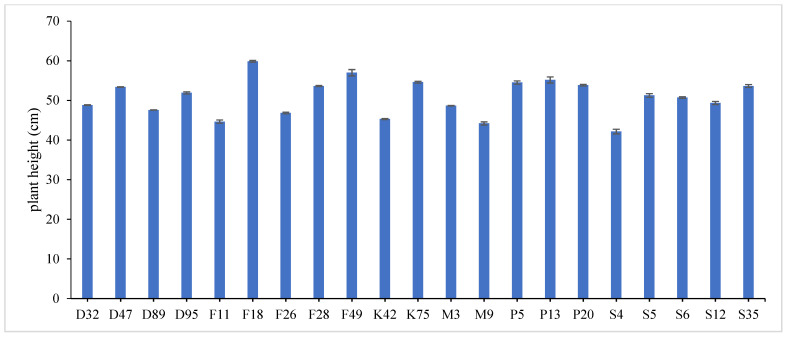
Mean performance of plant height, of twenty-one alfalfa genotypes at combined across the two years. Bar represents the standard deviation.

**Figure 2 genes-13-01521-f002:**
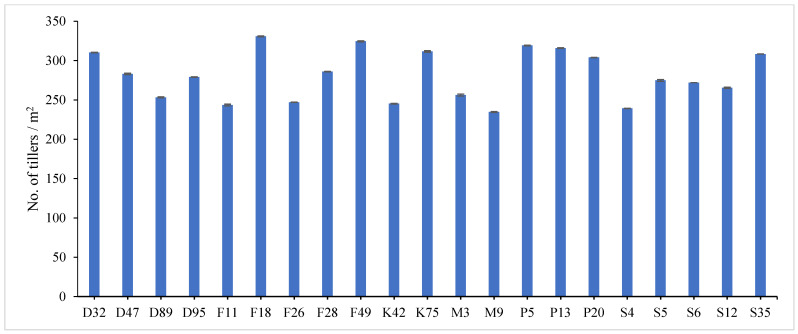
Mean performance of number of tiller/m^2^ of twenty-one alfalfa genotypes at combined across the two years. Bar represents the standard deviation.

**Figure 3 genes-13-01521-f003:**
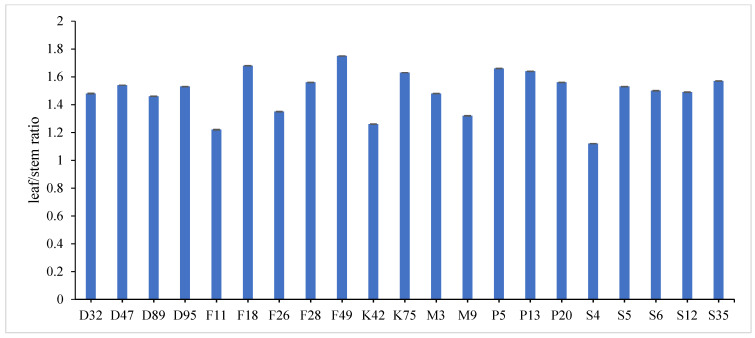
Mean performance of leaf /stem ratio of twenty-one alfalfa genotypes at combined across the two years. Bar represents the standard deviation.

**Figure 4 genes-13-01521-f004:**
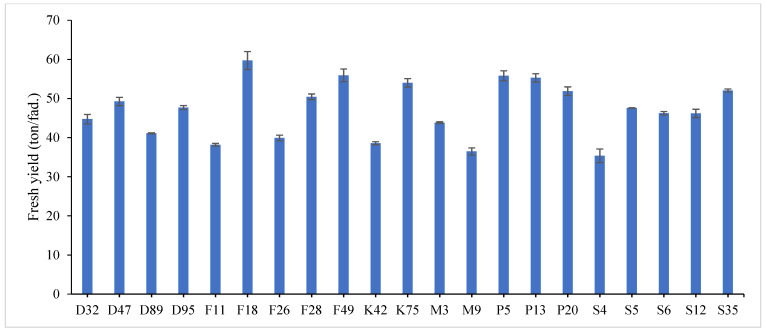
Mean performance of total fresh yield (ton/fed) of twenty-one alfalfa genotypes at combined across the two years. Bar represents the standard deviation.

**Figure 5 genes-13-01521-f005:**
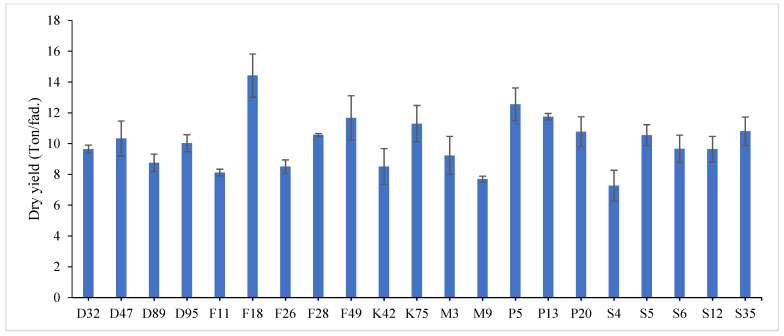
Mean performance of total dry yield (ton/fed) of twenty-one alfalfa genotypes at combined across the two years. Bar represents the standard deviation.

**Figure 6 genes-13-01521-f006:**
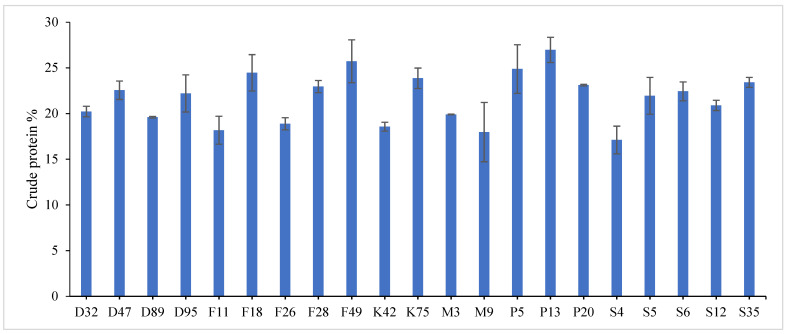
Mean performance of crude protein % of twenty-one alfalfa genotypes in the first year. Bar represents the standard deviation.

**Figure 7 genes-13-01521-f007:**
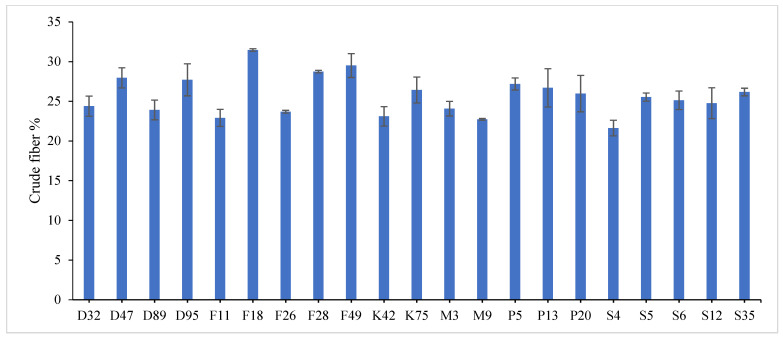
Mean performance of crude fiber % of twenty-one alfalfa genotypes in the first year. Bar represents the standard deviation.

**Figure 8 genes-13-01521-f008:**
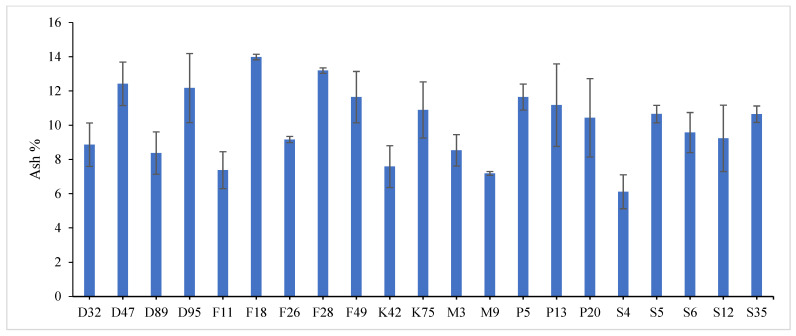
Mean performance of ash % of twenty-one alfalfa genotypes in the first year. Bar represents the standard deviation.

**Figure 9 genes-13-01521-f009:**
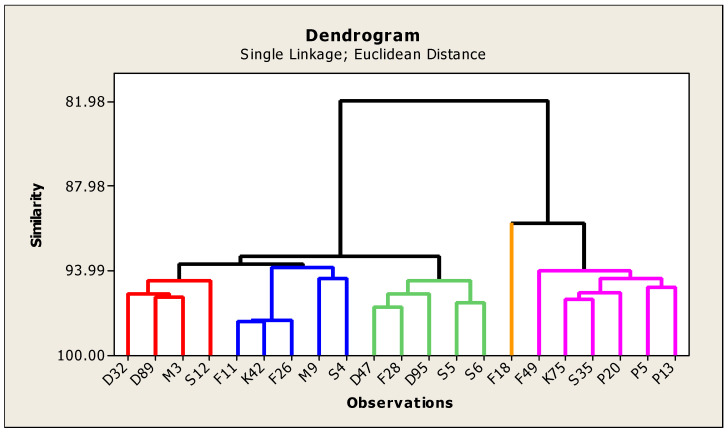
Dendrogram showing the distance among twenty-one genotypes of alfalfa based on yield components, forage yield and quality traits.

**Figure 10 genes-13-01521-f010:**
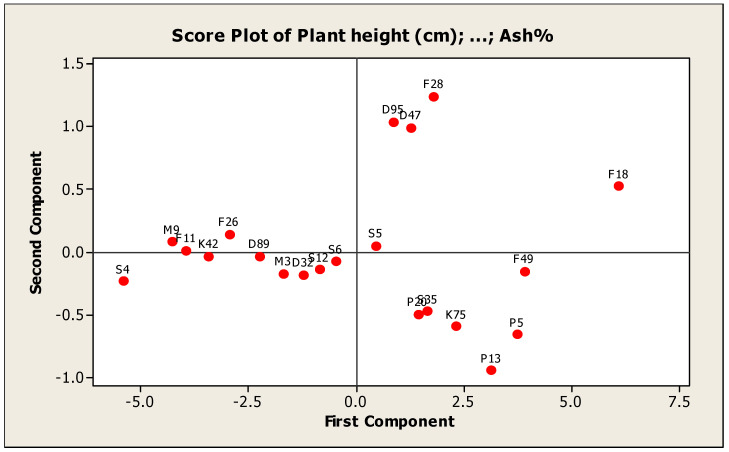
Principal component analysis of measured traits in twenty one genotypes of alfalfa.

**Table 1 genes-13-01521-t001:** List of alfalfa germplasm and collection site.

No.	Name	Collection Site from Egypt	No.	Name	Collection Site from Egypt
1	D32	New valley—El-dakhla	12	M3	New valley—Mout
2	D47	New valley—El-dakhla	13	M9	New valley—Mout
3	D89	New valley—El-dakhla	14	P5	New valley—Paris
4	D95	New valley—El-dakhla	15	P13	New valley—Paris
5	F11	New valley—Al-farafra	16	P20	New valley—Paris
6	F18	New valley—Al-farafra	17	S4	Matrouh—Siwa
7	F26	New valley—Al-farafra	18	S5	Matrouh—Siwa
8	F28	New valley—Al-farafra	19	S6	Matrouh—Siwa
9	F49	New valley—Al-farafra	20	S12	Matrouh—Siwa
10	K42	New valley—El-kharga	21	S35	Matrouh—Siwa
11	K75	New valley—El-kharga			

**Table 2 genes-13-01521-t002:** Soil mechanical analysis at EL- Ismailia agricultural station.

Soil Depth (cm)	Soil Characteristics				
	Particle Size Distribution				
	Coarse Sand	Fine Sand	Silt	Clay	Texture
0–15	68.00	25.75	3.82	2.46	Sandy
15–30	73.32	23.07	3.11	1.50	Sandy
30–45	75.20	20.97	3.00	0.83	Sandy
45–60	87.44	8.46	3.65	0.45	Sandy

**Table 3 genes-13-01521-t003:** Mean squares of yield components and forage yield of twenty-one alfalfa genotypes in two years.

Source of Variance	D.F.	Mean Squares									
		Plant Height (cm)		Number of Tillers/m^2^		Leaf/Stem Ratio		Total Fresh Yield (ton/fed)		Total Dry Yield (ton/fed)	
		First Year	Second Year	First Year	Second Year	First Year	Second Year	First Year	Second Year	First Year	Second Year
Reps	2	10.35	8.15	0.654	0.317	3.26	2.38	6.58	4.21	1.56	0.93
Genotypes	20	51.68 *	48.54 **	8.543 **	5.868 *	39.71 *	35.79 **	71.73 **	69.67 **	8.83 **	5.42 **
Error	40	1.26	1.19	0.39	0.28	1.14	1.08	1.18	1.09	0.31	0.24

* Significant at 5% level of probability, ** Significant at 1% level of probability.

**Table 4 genes-13-01521-t004:** Combined analysis of yield components and forage yield of twenty-one alfalfa genotypes across the two years.

Source of Variance	D. F	Mean Square				
		Plant Height (cm)	Number of Tillers/m^2^	Leaf/Stem Ratio	Total Fresh Yield (ton/fed)	Total Dry Yield (ton/fed)
Year (Y)	1	5.13	9.67	2.06	10.13	8.13
Error	4	1.86	1.04	0.08	1.37	3.88
Genotypes (G)	20	125.15 *	114.94 **	0.294 **	187.98 *	14.88 **
Y × G	20	8.34 **	25.22 **	0.056 **	18.17 **	7.87 *
Error	80	2.78	19.84	0.012	6.04	6.94

* Significant at 5% level of probability, ** Significant at 1% level of probability.

**Table 5 genes-13-01521-t005:** The percentage of variability parameters for agronomic studied traits in twenty-one alfalfa genotypes in the two seasons and combined across the two years.

Traits	Genotypic Coefficient of Variation (G.C.V%)			Phenotypic Coefficient of Variation (P.C.V%)			Broad Sense Heritability (H^2^_b_)		
	First Year	Second Year	Combined	First Year	Second Year	Combined	First Year	Second Year	Combined
Plant height (cm)	7.87	8.06	8.91	8.16	8.36	9.31	93.03	92.99	95.70
Number of Tillers/m^2^	0.58	0.50	1.43	0.62	0.54	1.51	87.45	86.93	94.70
leaf/stem ratio	22.26	24.10	14.53	23.03	25.14	16.69	93.42	91.89	87.06
Total fresh yield (ton/fed)	10.09	10.35	11.67	10.35	10.63	12.43	95.22	95.45	93.89
Total dry yield (ton/fed)	15.48	14.19	10.64	16.29	15.14	11.82	90.16	87.80	90.02

**Table 6 genes-13-01521-t006:** Mean squares of quality traits of twenty-one genotypes of alfalfa in the first year.

S.O.V.	D.f	Mean Squares		
		Crude Protein%	Crude Fiber%	Ash%
Rep.	2	9.96	28.17	2.07
Genotypes	20	28.67 **	107.54 *	11.86 **
Error	40	2.43	6.25	0.54

** Significant at 1% level of probability, * Significant at 5% level of probability.

**Table 7 genes-13-01521-t007:** The percentage of variability parameters for quality of studied traits in twenty-one alfalfa genotypes in the first year.

Traits	Genotypic Coefficient of Variation (G.C.V%)	Phenotypic Coefficient of Variation (P.C.V%)	Broad Sense Heritability (H^2^_b_)
Crude protein%	12.52	14.15	88.48
Crude Fiber%	21.64	23.56	91.85
Ash%	20.04	21.42	93.56

**Table 8 genes-13-01521-t008:** Summary of cluster analysis shows the twenty-one genotypes of alfalfa.

Cluster No.	No. of Genotypes in Each Group	Included Genotypes
1	4	D32, D89, M3 and S12
2	5	F11, K42, F26, M9 and S4
3	5	D47, F28, D95, S5 and S6
4	1	F18
5	6	(F49, K75, S35, P20, P5 and P13

**Table 9 genes-13-01521-t009:** Principal component analysis of measured characteristics in twenty-one genotypes of alfalfa.

	PC1	PC2	PC3	PC4	PC5
Plant height (cm)	0.325	0.037	0.125	0.026	−0.223
Number of tiller/m^2^	0.317	−0.322	0.037	0.472	−0.457
Leaf/stem ratio	0.309	−0.104	0.527	−0.726	−0.114
Total fresh yield	0.331	−0.178	0.052	0.051	0.013
Total dry yield	0.321	−0.105	−0.411	−0.150	0.117
Crude protein%	0.312	−0.288	0.424	0.383	0.447
Crude Fiber%	0.307	0.584	−0.013	0.103	−0.496
Ash%	0.304	0.629	0.113	0.140	0.489
Eigen value	5.76	3.11	1.53	1.34	1.15
Variation Explained (%)	53.24	15.73	13.21	7.93	5.87
Cumulative Variance (%)	53.24	68.23	74.32	85.65	88.50

## Data Availability

All data generated during this study is presented in the manuscript and Appendix A.

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
