# Peer review of "Genetic Diversity, Analysis of Some Agro-Morphological and Quality Traits and Utilization of Plant Resources of Alfalfa"

_genes, 2022, doi:10.3390/genes13091521_

Round 1

Reviewer 1 Report

The article is written in a logical and orderly manner. It contains a lot of interesting information.

The manuscript can be accepted in its current version.

Reviewer 2 Report

The manuscript genes-1860544, entitled “Genetic Diversity, Genetic Analysis of some Agro-Morphological and Quality Traits and Utilizing of Plant Genetic Resources of Alfalfa” submitted by Sayed et al. reported and discussed the results of a two-year field experiment set up in Egypt where the morphological and genetical diversity was assessed on 21 different alfalfa genotypes.

In general, the experimental activity was carried out following a strict scientific logic and according to widely used methods however, the manuscript’s quality is very poor and It must be revised on in all its parts. In particular, language must be revised by a native speaker or from a proofediting service. Indeed, many sentences are difficult to read and understand.

Abstract: must be rewritten in many parts and I suggest to the authors to add more quantitative information.

Introduction: in general it is good.

Materials and Methods: revise it according to my suggestion enclosed in the attached file.

Results and discussion: results are clear although some sentences should be rewritten. Discussion is very poor and limited only to some sentences/references. The Graphs must be revised by adding bars and statistical analysis. Statistical differences among genotypes were not presented.

Conclusions are clear and summarize the main results observed in the experiment.

My specific comments, which I hope will help the authors to improve their manuscript, are enclosed in the attached pdf file.

Round 2

Reviewer 2 Report

The authors applied almost all suggested changes with a significant improvement of manuscript qualitu. Therefore, in my opinion, I suggest to accept the article for publication.